# 3D Cell Culture of Human Salivary Glands Using Nature-Inspired Functional Biomaterials: The Egg Yolk Plasma and Egg White

**DOI:** 10.3390/ma13214807

**Published:** 2020-10-28

**Authors:** André M. Charbonneau, Simon D. Tran

**Affiliations:** McGill Craniofacial Tissue Engineering and Stem Cells Laboratory, Faculty of Dentistry, McGill University, Montréal, QC H3A 2B2, Canada; andre.charbonneau2@mail.mcgill.ca

**Keywords:** egg yolk, egg yolk plasma, egg white, three-dimensional cell culture, multi-compositional, biomaterial, salivary gland, tissue engineering

## Abstract

The egg yolk plasma (EYP)—a translucent fraction of the egg yolk (EY) obtained by centrifugation—was tested as a developmentally encouraging, cost-effective, biomaterial for salivary gland (SG) tissue engineering. To find optimal incubating conditions for both the human NS-SV-AC SG acinar cell line and SG fibroblasts, cells were stained with Live/Dead^®^. The cellular contents of 96-well plates were analyzed by high content screening image analysis. Characteristically, the EYP biomaterial had lipid and protein content resembling the EY. On its own, the EYP was non-conducive to cell survival. EYP’s pH of 6 mainly contributed to cell death. This was demonstrated by titrating EYP’s pH with different concentrations of either commercial cell culture media, NaOH, or egg white (EW). These additives improved SG mesenchymal and epithelial cell survival. The best combinations were EYP diluted with (1) 70% commercial medium, (2) 0.02 M NaOH, or (3) 50% EW. Importantly, commercial medium-free growth was obtained with EYP + NaOH or EYP + EW. Furthermore, 3D cultures were obtained as a result of EW’s gelatinous properties. Here, the isolation, characterization, and optimization of three EYP-based biomaterial combinations are shown; two were free of commercial medium or supplements and supported both SG cells’ survival.

## 1. Introduction

The acini of salivary glands (SG) are functional clusters of cells that produce saliva. They can be severely damaged from head and neck cancer’s irradiation therapy, Sjogren’s autoimmune syndrome, or as a result of medication side effects [1,2]. Without available treatment to restore permanent salivary flow, tissue engineers want to produce implantable miniature salivary secretory units [3,4,5,6,7]. Furthermore, another benefit of engineering SGs—or tissues in general—are their implementation as in vitro models that are usable in preclinical studies [8,9].

In tissue engineering, biomaterials possess a central role; they constitute the cells’ environment. Accordingly, the biomaterials should provide a controlled physiologically mimicking environment and tissue growth promoting cues [10,11,12,13]. For soft tissues such as SGs, there are soft biomaterials. Some biomaterials are composed of a few elements. They are inexpensive, abundant, highly controllable, and can usually be purified from natural sources or synthesized with a high-degree of monodispersity [14,15,16,17,18]. Alternatively, biomaterials can consist of multiple elements. With multiple elements, these biomaterials recapitulate to a greater extent the biochemical complexity found in the tissue microenvironment [19,20,21,22,23]. Multi-compositional biomaterials can require extensive decellularization, be tumor-derived, or be scarce [10]. In an attempt to create a low cost and abundant alternative multi-compositional functional biomaterial, the chicken’s egg was selected as a starting material. With the successful growth and survival of SG cells, other glandular organs such as lungs and pancreas could be tested.

On a weight-to-weight basis, the egg yolk (EY) contains more proteins and lipids than the egg white (EW) [24], suggesting its superiority as a biomaterial candidate. Our pilot studies showed some limitation regarding its opacity affecting cell visibility during microscopy. Interestingly, EY has been lightened by dilution and centrifugation by many food scientists [25]. With recent food science studies diluting EY before centrifugation, the centrifuged EY becomes (1) the supernatant translucent fraction—which we refer to as diluted egg yolk plasma (**_D_**EYP), and (2) the opaque pellet—which we refer to as diluted granules (**_D_**GRA). _D_EYP on a dry weight contains ≈ 75% lipids (mainly low-density lipoproteins (LDL)) and ≈ 25% proteins [25,26,27,28]—mostly globular proteins named “livetins” derived from the hen’s serum [28,29]. Seeing as food scientists diluted the EY, we felt it would be important to maintain as much of the original content, similar to what cells would encounter during embryonic development. Converting the EY into a translucent form without dilutions could fit the multi-compositional, functional, low-cost, and abundant biomaterial criteria.

Avian cells’ interaction with egg biomaterials has been studied by chick embryologists for decades. The egg’s environment allows 60,000 cells to become an entire organism within 21 days ex vivo [30]. In some cases, modifications to the embryo’s in ovo environment [31,32,33] or removal from it [34,35,36,37,38,39,40] still permit the embryo’s development and survival. In some in vitro cell culture applications, scientists have also used blended chick embryos [39,41]. Fujii and Gospodarowicz grew bovine endothelial cells in medium supplemented with up to 20% EY. Their experiments showed how EY may lack cell substratum attachment molecules and noted how _D_EYP + EW’s or _D_EYP + EW + medium’s combinations did not permit cell growth [42]. Murakami et al. focused more specifically on _D_EYP and isolated a potent growth-promoting factor from this fraction. The authors used the factor “YLP-pI7.5” in attempts to grow plasmocytomas, epithelial, and fibroblasts cells in serum-free media [43]. More recently, studies have incorporated human embryonic stem cells into developing chick embryos [44,45] that matured into developing xenograft mammalian tissues. Scientists have also used the entire developing egg and its chorioallaentoic membrane (a vascular membrane found in eggs) as an angiogenic preclinical drug-testing device [9,46]. Recently, the usage of egg-derived biomaterials as a 3D gel (especially the EW [47,48]) and most recently in EY and its fractions [49] have been investigated. Many questions remain, since few studies have investigated human cell survival in egg-derived materials. Previous studies of human cells in egg biomaterials have either diluted the egg biomaterials (_D_EYP, EW) with a culture medium or worked with fertilized eggs. However, it remains unknown if human cells can survive in unfertilized eggs without cell culture medium.

Since avian and human cells can survive in the egg during chick development, we hypothesized that they could survive in vitro in non-developing, translucent, non-diluted EYP without commercial supplements or culture medium. To tissue engineer salivary secretory units and test our novel biomaterial, two SG cell types were seeded in the egg biomaterial. The cell types included (1) the NS-SV-AC acinar cell line and (2) primary human SG fibroblasts (HuSG-Fibro). High-throughput screening was chosen to evaluate the cells’ survival and dynamics.

This study describes the isolation of the EYP and the appropriate modifications it received to permit human cell survival in vitro. Importantly, under optimized condition, cells survived free of commercial supplements and medium and when mixed with EW permitted spheroid formation in a 3D distribution.

## 2. Materials and Methods

### 2.1. Isolation and Preparation of Egg Yolk Plasma (EYP) and Egg White (EW) for Cell Culture

Chicken egg dozens (Large White Eggs Omega 3, Presidents’ Choice) were purchased from local grocery stores (Montréal, Canada), selected with the furthest expiry date, and used within the first week and never beyond expiry. Eggs were produced from *Gallus domesticus*. Prior to isolating the egg’s inner content in a biosafety cabinet, the shells’ outer surfaces were sprayed with 70% ethanol. On a 250 mL sterilized beaker’s edge, eggs were manually cracked, and the EW was drained while the EY was maintained in the shell. EW homogenization was accomplished by vacuum aspiration through a large porous funnel (60240, Coors, Golden, CO, USA) three times. For short-term storage, the filtered EWs were stored in a sterilized 50 mL tube. Separated from the EW in the biosafety cabinet, EYs were individually poured out onto a 70% ethanol sprayed and air-dried paper towel. EYs were rolled on the paper towel to remove the remaining EW until they appeared dry. From the paper towel, we individually added eggs (to a maximum of 4 eggs) with vitellin membrane to a pre-sterilized 250 mL beaker. With a metal spatula, EYs were mixed with 10–15 swirls. To transfer the homogenized EYs into many 2 mL tubes for centrifugation, we used a 1 mL pipette with tips clipped at the apex. The 2 mL tubes containing the EYs were centrifuged (5417R, Eppendorf or 22KBR, Marathon, Pittsburg, PA, USA) for 6 h at 15,400 g at 25 °C. Following the spin, all the supernatant EYP (Figure 1A) was collected into 50 mL tubes and kept for storage at 4 °C until pasteurization. For experiments using the historically described _D_EYP, various preparation methods exist [25,26,50,51]. Our protocol resembled Guilminueau et al.’s [26], but also included the EY’s membrane. Briefly, we filled a 2 mL tube with EY diluted at 1:1 with 0.17 M NaCl. The samples were spun for 45 min at 10,000 g at 10 °C. For pasteurization, 50 mL tubes containing either _D_EYP, EYP, or EW were heated with a water bath (Isotemp 205, Fisher Scientific, Waltham, MA, USA) until a thermometer in the biomaterial indicated 57–61 °C for 5 min. After their heating, we cooled the biomaterials at −80 °C until they froze (≈ 20 min) in a −80 °C freezer. After freezing, the egg biomaterials were thawed at 37 °C for immediate usage or placed at 4 °C for storage. The purchased eggs were catalogued through purchase, processing, and usage.

### 2.2. Protein Quantification

Due to a high protein content in the EYP samples, several dilutions were required for analysis with Bicinchoninic Acid (BCA) Kit (23225, Pierce, Rockford, IL, USA). For the analysis of EY, EYP, _D_EYP, and GRA samples centrifuged at T = 0, 2, 4, and 6 h (Figure 1B) and EYP + Medium (Figure A1 in Appendix B), they were serially diluted with 25% glycerol twice to 0.6% (*v*/*v*). Then, 18 μL of 0.6% egg biomaterials reacted with 120 μL of the BCA kit reagents. With 100% cell culture media samples, a protein isolation column (9K MWCO 89884A, Pierce, Rockford, IL, USA) was used to concentrate the proteins. Briefly, 7 mL of both medium were placed in separate columns and spun at 2500g for 40 min at 22 °C. EpiMax medium ^®^ (002010024 CL, Wisent, St-Bruno, QC, Canada) yielded ≈ 200 μL that was diluted with 25% glycerol to 16% (*v*/*v*). For the Complete Growth Medium (see cell culture section for detail), the spin yielded 250 μL, and this was diluted to 6% (*v*/*v*). For the media proteins to react with the BCA kit reagents, 18 μL from the dilutions—either the 16% or 6%—were transferred into test vessels having 168 μL final volume. These quantitative experiments had triplicates, a standard curve with bovine serum albumin (BSA), were conducted in 96-well plates and analyzed on a plate reader (ELx 800, BioTech, Winooski, VT, USA).

### 2.3. SDS-PAGE and Its Staining with Coomassie and Suddan Black B

Tris-Cl acrylamide gels of 5 and 15% were cast following the guidelines from Current Protocols in Molecular Biology [52]. Reagents were from the following sources, Tris Base (BP152-1, Fischer Scientific), 30% acrylamide/0.8% bisacrylamide (161-0158, BioRad, Mississauga, ON, Canada), Glycine (161-0718, BioRad), SDS (161-0302, BioRad), and molecular weight marker (161-0374, BioRad). Prior to loading 5 μg per lane in a 15-lane gel, proteins were pre-heated to 56 °C for 1 h and at 100 °C for 5 min. Voltages for stacking and running gels were of 70 V and 100 V, respectively. Run gels were stained with Coomassie Stain (161-0803, BioRad) by following the manufacturer’s protocol. For Suddan Black B, Prat’s protocol [53] was followed. Briefly, 0.25 g of Suddan Black B (199664, Sigma-Aldrich) was dissolved in 10 mL acetone, 7.5 mL acetic acid, and topped up with water to a final volume of 50 mL. The solution was spun at 350 g for 5 mins. The black liquid was incubated with the gel overnight. Suddan Black B excess stain was removed with three, 3–5 h rinses. The rinsing liquid was composed of acetic acid 15%, acetone 20%, and water. The gels were digitized with a paper scanner.

### 2.4. HuSG-Fibro and NS-SV-AC Cell Culture

Non-malignant human submandibular gland tissues were surgically procured by the Oral and Maxillofacial Surgery department at McGill University Hospitals in compliance to guidelines from the McGill Research Ethics and Compliance Internal Review Board. The cells specifically shown in the experiments were from a 27-year-old female, while other samples received were from patients with a mean age of 43 that ranged between 20 and 74. The isolation of HuSG-Fibro followed the alternative protocol in Current Protocols in Cell Biology named, “Establishment of Fibroblast Cultures” [54]. Briefly, human salivary tissues were minced with scalpels and incubated with 1000 U/mL of collagenase Type 1 (LS004196, Worthington, Lakewood, NJ, USA) at 37 °C for 2 h. HuSG-Fibro cells were grown in RPMI 1640 based Complete Growth Medium (21870076, ThermoFisher Scientific, Waltham, MA, USA) including, 10% fetal bovine serum (FBS), 1% 1 M HEPES, 1% non-essential amino acids, 1% L-glutamine, 1%, penicillin/streptomycin, and 1% sodium pyruvate. The cells of interest attached overnight while non-adherent cells and erythrocytes were removed in the subsequent medium change. Acinar cell line NS-SV-AC (created by Professor Masayuki Azuma, Tokushima University) [55,56] were expanded in EpiMax^®^ medium (002010024 CL, Wisent, St-Bruno, QC, Canada). The EpiMax contained supplements: hormones and Bovine Pituitary Extract (002013024, Wisent), Supplement Antibiotic (002014016 IL, Wisent) and Supplement Gentamycin (002015017 TL, Wisent). The expansion occurred in T25 flasks, 6-well plates, or 100 mm dishes. 0.05% Trypsin/ EDTA was used to passage both cell types. The histologic characterization of both cell types grown on glass was conducted with chemical stain Sirius Red, Alcian Blue + PAS, and by immunohistochemistry using antibody mixture, Collagen Panel (PA1-36058, ThermoFisher Scientific, Waltham, MA, USA) (Figure A2).

### 2.5. Culture of HuSG-Fibro and NS-SV-AC Cells in Biomaterials

NS-SV-AC cell line was thawed and passaged once for expansion prior to use, while HuSG-Fibro cells were expanded and used between passages 2 and 5. From their medium cultures, the cells were condensed by centrifugation and resuspended in volumes <500 μL for distribution into biomaterial mixtures at appropriate density. Experimental control “biomaterials” included autoclaved and syringe filtered (0.22 μm) phosphate-buffed saline (PBS) and PBS diluted BSA (A3803, Sigma, St. Louis, MO, USA) to 60 mg/mL. For egg-based biomaterial preparation, see the Materials and Methods sections: “Isolation and Preparation of EYP and EW for Cell Culture” and “pH Measurements and pH Modification of EYP” below. Condensed, isolated cells were transferred into a 15 mL tube containing the biomaterials. To mix the cells into the more viscous egg biomaterials, the 15 mL tubes were temporarily incubated at 37 °C and inverted 15 times. A 1 mL syringe (309624, BD Bioscience, San Jose, CA, USA) equipped with a 21-gauge needle was used to transfer the cell laden biomaterial from the 15 mL tube to a 96-well plate (6005225, Perkin Elmer, Waltham, MA, USA). Each well of the plate contained approximately 100 μL of biomaterial and 2500 cells per well. After initial seeding, at T = 0 and on every 4th day, 50 μL of biomaterials were added without cells.

### 2.6. Live/Dead Staining, Imaging and Analysis

Calcein AM (Ab 141420, Abcam, Toronto, ON, Canada), DMSO, Ethidium Homodimer (EthD) (L3224, Invitrogen, Waltham, MA, USA) and cell culture medium (EpiMax or Complete Growth Medium) were combined into a master mix. The master mix’s final volume was calculated to add 50 μL/well. The final in well concentrations of the Calcein AM and EthD dyes were 0.125 μM and 0.5 μM, respectively. With the exception of EYP + EW, a visible fluorescent signal appeared within 1 h. For best results, the gelatinous EYP + EW mixture required a longer incubation period (≈3 h) and gentle mechanical pipette piston actuations to mix the dyes deeper. A microscope with bright field and fluorescent capabilities containing a 10 and 20X objectives was used for non-quantitative imaging (DMIL, Leica, Concord, ON, Canada). For quantitative imaging, a 2X objective on Perkin Elmer’s Operetta high content screening microscope was used. Image analysis algorithms were constructed with Volocity 5.4.1 (Perkin Elmer) (Figure A3). The signal to noise thresholds for quantifications were standardized by setting the threshold to a number of positive standard deviations beyond the maximum peak in a pixel value frequency plot (Figure A3C). Size exclusion filters were only applied with objects at T = 0 stained with Calcein AM and with all EthD images. The term “Total sum of surface area” quantified the total surface area (2D) in μm^2^ of all the Calcein AM positive cells in a single well. The term “Dead cell counts” quantified the number of EthD nuclei distinguishable from others in 2D. The term “Individual object surface” looked at each individual Calcein AM cell or cluster that can be distinguished from others in proximity and calculated—in 2D—their average surface area in μm^2^ per well.

### 2.7. pH Measurements and pH Modification of Egg Yolk Plasma (EYP)

Five mL of EY, EYP, EYP + NaOH, EW, medium, or combinations of these mixtures were placed in 15 mL tubes. For mixing, tubes were inverted 10 times. pH meter (UB-10, Denver Instrument) was used to test the samples in a 37 °C water bath. We obtained an EYP of pH 7.4 by using 0.25 mL of 0.375 M NaOH (0.2 μm filtered) in 5 mL of EYP. NaOH was distributed through the EYP with inversions. For the experiment adding PBS to EYP pH 7.4, EYP’s NaOH molar concentration of 0.0188M was calculated before the addition of PBS. In other words, there were identical initial volumes for the EYP 7.4 preparations (with/without dilution), but after dilutions, the NaOH molar concentration changed according to the percentage (*v*/*v*) of PBS in the final volume. In cell culture experiments, pH measurements were taken with pH paper (109533, Milipore Sigma, Billerica, MA, USA) at the start and end of the culture periods (Figure A6).

### 2.8. Statistical Analysis

GraphPad Prism version 7 (GraphPad Software) was used to perform the statistical analysis. Most data were presented with mean ± S.E. and when appropriate were analyzed by one-way ANOVA followed by Tukey’s Post-Hoc test. The statistical significance was defined when p-value < 0.05.

## 3. Results

### 3.1. Isolation and Characteristics of Egg Yolk Plasma (EYP)

EY’s initial average protein concentration was of $\overset{\ldots }{\bar{x}}$ = 95 mg/mL (SE ± 9 mg/mL) (Figure 1B). After 2 h of centrifugation, the majority of the proteins were separated (Figure 1B), but the supernatant was still turbid (Figure 1A). Then, only after 6 h of centrifugation was the separation of the undiluted EY visually distinguishable into two distinct fractions: (1) now a clear translucent EYP supernatant and (2) an opaque GRA pellet (Figure 1A). The clear translucent supernatant obtained after 6 h of spinning was our desired biomaterial, the EYP. The centrifugation of 3–4 EYs usually yielded 35 mL of EYP. The centrifugation time affected the protein concentration in different fractions. After 6 h centrifugation, the mean protein concentration in the EYP was $\overset{\ldots }{\bar{x}}$ = 72 mg/mL (SE ± 3 mg/mL), while in the GRA fraction, it was $\overset{\ldots }{\bar{x}}$ = 132 mg/mL, (SE ± 9 mg/mL). The mean EYP concentration remained substantially above mean _D_EYP levels ($\overset{\ldots }{\bar{x}}$ = 24 mg/mL, SE ± 1 mg/mL) (Figure 1B) or cell culture media levels (0.3–1 mg/mL) (Figure A1). SDS-PAGE Coomassie stained gels showed similar protein molecular weights between EYP, non-pasteurized EYP, and _D_EYP (Figure 1C,D). In these gel lanes, 26 bands were detected: 15 above and 11 below 75kDa. Unspun EY had two extra bands near 110 kDa and 31 kDa. As a means of comparison, all egg biomaterials had more bands and a greater overall molecular weight band distribution than both media (EpiMax and Complete Growth Medium). Similarly, the gel bands stained by the Suddan Black B stained—to a greater extent—egg-based biomaterials as compared to both media (Figure 1D).

### 3.2. In EYP, NS-SV-AC, or HuSG-Fibro Required at Least 30% Commercial Medium for Cell Survival

When initially seeded in the EYP, NS-SV-AC or HuSG-Fibro cells fell to the base of the well and all stain Calcein AM (live) positive (Figure 2G). Over time, cells became EthD positive (dead) (Figure A5B,M). Importantly, the results went against our initial hypothesis. In attempts to improve survival, we diluted EYP with commercial medium respective of each cell type’s regular medium requirements; accordingly, serum-free EpiMax for NS-SV-AC and serum-supplemented RPMI 1640 for HuSG-Fibro cells. With both cell types, a dilution of 30% medium or greater increased or maintained the Calcein AM surface area. In contrast, EYP diluted with 30% PBS—used as a control—showed no cell survival similar to EYP and EYP + 10% medium (Figure 2H,I). PBS did not match the cell culture medium’s ability to rescue cell survival.

More specifically for the NS-SV-AC, medium concentrations showed a significant improvement from 30% to the 50–90% range (Figure 2A,J,K). Seventy percent medium brought the greatest increase in total Calcein AM surface area (Figure 2A) and showed significantly the least EthD-positive cells (Figure 2C). Cell attachment properties were also modified over time. Without medium, cells remained alone (Figure A5B). With 30% medium, we noted that objects started clustering (Figure 2J). Between 50 and 90% medium, we observed no significant difference in individual clusters’ surface area (Figure 2E). Bright field imaging revealed darker shades of cells’ cytoplasm dependent on dilution (50 and 70 vs. 90% dilution) (Figure A4C–F). For seeding density, increasing cell numbers caused larger clusters, but above 3000 cells/well, there were not more clusters (Figure A7). Importantly, for the NS-SV-AC, the EYP permitted the formation of clusters (sphere-like structures) at the base of the dish contrasting its sheet conformation in conventional pure EpiMax medium (Figure A5K).

More specifically, for the HuSG-Fibro, they were also receptive to the addition of commercial medium starting at 30% (Figure 2L–N). In this cell type, dead cell quantities always rose in the first days but stabilized (Figure 2D). In certain conditions, total Calcein AM surface area increased (Figure 2B). We observed the lowest EthD nuclei counts in mixtures having equal or greater than 50% medium (Figure 2D). Morphologically, cells in 30–50% medium (inclusively) formed small cell clusters (individual objects) at the base of the dish; above these culture medium dilutions, cells tended toward a monolayer conformation (Figure 2L–N and Figure A4J–N). The largest individual objects were detected in 70% medium (Figure 2F). Similar to the NS-SV-AC, higher EYP concentrations affected—in a dose-dependent manner—cytoplasmic light transmission (Figure A4J–N). In bright field microscopy, we also observed—in condition with ≥ EYP + 50% medium—a substance around the cell clusters. The substance was voided of both Calcein AM and EthD fluorescence (dotted outline; Figure 2M). In EYP, the HuSG-Fibro did not proliferate at the same speeds as in medium but may have been producing a substance.

Importantly, EYP alone was not conducive to cell survival. For survival to occur, both cell types required EYP to be supplemented with at least 30% medium.

### 3.3. Culture Medium Modified EYP’s pH; so Did NaOH and EW

The addition of a commercial medium to EYP was conducive to cell survival. Furthermore, at this point, EY’s pH was identified from the literature to be at 6. In an attempt to consolidate these findings, pH measurements were conducted on EYP and EYP + EpiMax mixtures. EpiMax modified EYP’s pH in a dose-dependent manner and to a greater extent than PBS (Figure 3A). EYP’s pH could also be modified using other cost-effective substances. For instance, 0.25 mL of 0.375M NaOH could modify 5 mL of EYP from pH 6.06 (S.D ± 0.062) to pH 7.37 (S.D ± 0.14) (Figure 3B). Interestingly, the same dosage did not modify EY’s pH to the same extent 6.69 (S.D ± 0.09) (Figure 3B). Most importantly, at 50%, EW influenced EY’s and EYP’s pH from near 6.0 to 7.11 (S.D ± 0.12) and 7.32 (S.D ± 0.08), respectively (Figure 3B). The homogenized EW’s pH was 8.75 (S.D ± 0.091). With EYP’s and EW’s pH found on opposing ends of the pH scale, both NaOH or EW could bring a dose-dependent modification to EYP’s pH.

Importantly, with NaOH and EW modifying EYP’s pH, it was hypothesized that these additives could alleviate EYP’s requirement for commercial culture medium and hormone supplements; therefore, leading us closer to finding our cost-effective abundant, functional, multi-compositional biomaterial.

### 3.4. In EYP + NaOH (EYP 7.4) NS-SV-AC and HuSG-Fibro Survived without Commercial Medium

In contrast to EYP alone, EYP 7.4 permitted both NS-SV-AC and HuSG-Fibro’s survival without cell culture medium (Figure 4). Over the course of the experiment, it was observed that EYP or EYP 7.4 dried when exposed to incubating conditions (data not shown). EYP 7.4 would benefit from hydration. PBS was used as a hydrating agent in a dose-dependent manner to improve EYP7.4 culture’s outcome (Figure 4).

With the NS-SV-AC, EYP 7.4 with 30–70% PBS had significantly similarly the best fold increase in Calcein AM total surface area (EYP 7.4 + 30% PBS = 2.75, EYP 7.4 + 50% PBS = 2.61, EYP 7.4 + 70% PBS = 2.69 (Figure 4A). For EthD-positive nuclei counts, 100% medium still had significantly the lowest fold increase after 14 days (EYP 7.4 + 30% PBS 15.5, EYP 7.4 + 50% PBS 13.2, EYP 7.4 + 70% PBS 12.8 vs. medium 7.47) (Figure 4C). As controls, PBS and BSA 60 mg/mL performed similar to pure EYP showing rapid generalized death within 4 days. Although BSA was infused in PBS with similar protein quantities as EYP + 50% dilution, it did not promote cells’ survival. For individual object’s surface area, EYP 7.4 + 50% PBS had significantly the largest mean individual object surface area fold increase over 14 days (EYP 7.4 + 50% PBS 10.4 fold) (Figure 4E). In terms of morphology, the living cells showed more often sheets rather than clusters (Figure 4K,L and Figure A5C–H). Cytoplasmic light transmittance was most affected by higher EYP concentration as compared to medium, BSA, or PBS (Figure A5C–H).

As for HuSG-Fibro in EYP 7.4 + PBS, the survival trend followed our EYP + Medium’s observation. Accordingly, the total Calcein AM surface area first slightly decreased and then stabilized. Increases in surface area were only observed in medium positive control condition (medium 12.0 fold vs. others 0.24–0.59 fold) (Figure 4B). For proliferation, this cell type appeared to depend to a greater extent on components of the commercial medium. EYP 7.4 + 50%PBS had the largest individual Calcein AM surface area after 14 days (Figure 4D). For EthD-positive nuclei, all conditions showed relatively lower dead nuclei counts (4.78–13.7 fold) as compared to BSA, PBS, and EYP (≈35–42 fold) (Figure 4D). Morphologically, all EYP 7.4 + PBS mixtures did not allow cell spreading similar to the medium (Figure A5O–S vs. V). In bright field, EYP caused darker cells’ cytoplasm (Figure A5N–S,V), and the same non-fluorescent substance as seen with EYP + Medium mixture was observed here near the cell clusters (Figure 4N). HuSG-Fibro cells appeared more productive than proliferative.

Importantly, these experiments showed how NaOH and PBS concentrations can permit both cell survival without commercial medium and that both cells perform best with 50% PBS hydration. In the culture wells, pHs did not change over 14 days (Figure A6), suggesting that the egg biomaterials’ pHs remain relatively stable over time.

### 3.5. When EYP was Mixed with Egg White (No Commercial Medium or Supplements), Salivary Cells Proliferated and Form Spheroids in 3D

EYP’s pH could also be modified by EW (Figure 3B). EYP + EW biomaterial combination was benchmarked for cell survival and 3D distribution of cells (Figure 5). At T = 0, 3D distributed cells were generally Calcein AM positive. Early on, NS-SV-AC cells rapidly expanded from ≈ 5 μm single cells to multi-cell ≈ 50 μm size clusters. Beyond 4 days, NS-SV-AC clusters’ size remained relatively stable. As seen in the other experiments, the HuSG-Fibro cells were not as proliferative but visually showed the least dead cells. At most, near 7 days, the HuSG-Fibro had formed 2–4 cell clusters that were mostly Calcein AM positive. Bright field images showed once again both cell types appearing darker as time passed, and fluorescent images showed how the darker cells were also Calcein AM positive. Importantly, EYP + EW permitted both cell types survival without cell culture medium and showed clear 3D distribution (Appendix A) to at least 14 days.

## 4. Discussion

Initially to isolate EYP, food scientists used long centrifugation times [50,57] or ultracentrifugation [51]. Later, diluted EYP (_D_EYP) became more common [25,58], which was most likely as a result of food scientists’ desire to decrease centrifugation time and speeds. The new EYP isolation protocol contrasts these previous protocols in three main aspects: (1) The protocol did not require ultracentrifugation, rather only the widely available tabletop centrifuge. (2) The protocol included the vitellin membrane. Although it was ruptured in the process, it was included, since it is known to foster the avian blastoderm during development [59] and contains glycoproteins similar to those found in tissue extracellular matrix (ECM) [30]. (3) EYP was not diluted prior to centrifugation to maintain as much original content as possible. There appears to be a reason for such high protein concentration since decreases in egg content cause abnormalities during chick embryology [30,35,60,61,62,63]. Although EYP was not diluted prior to centrifugation, EYP + Medium or EYP7.4 + PBS biomaterials still required dilution for optimal cell survival. The dilution with medium or PBS decreased protein quantities. High protein concentrate biomaterials such as EYP + EW should best supply maximal nutrients for the development of larger tissue in vitro. Surpassing by far serum-free EpiMax (Figure A1 ≈ 0.3 mg/mL) and 10% serum supplemented complete growth medium (data now shown ≈ 1 mg/mL).

In terms of protein molecular weights, our data showed that _D_EYP and EYP display identical Coomassie stained SDS page gel patterns. Based on the SDS-PAGE gel pattern similarities between EYP and _D_EYP, there is a high probability that a majority of the proteins in the EYP can be deduced from published data on the _D_EYP [25,26]. In previous mass spectrometry studies, Mann and Mann were able to identify 100 _D_EYP proteins [27]. In the context of cell culture, the protein bands in EYP determined by SDS PAGE were compared to the literature and estimated to be mostly LDL apoproteins and livetins (serum albumin, immunoglobulin Y, and α2-glycoprotein). These proteins could contribute to amino acid requirement for new protein synthesis, act as anti-oxidants, transmembrane transporting vehicles, or a primitive immune system (similar to the humoral immune response). The literature that exists on egg protein’s function during avian development can be used to extrapolate the proteins’ roles in human cell culture applications [28,29,30]. When comparing EYP to commercial culture media, the media contained fewer lipids (Figure 1D), proteins, and protein molecular weight variability (Figure 1C). Once again, there appears to be a reason for such high protein concentration since a decreased in egg content causes abnormalities during chick embryology [30,35,60,61,62,63]. Furthermore, crowded environments—especially mixed molecular crowded environments—have been shown to increased cells’ ECM production [64,65]. EY and EYP gel band patterns slightly differed. EY’s bands near ≈ 110 kDA and 31 kDa (high-density lipoprotien [26]) absent from EYP were most likely displaced to the GRA pellet by centrifugation. As a consequence, they did not enter the cell culture experiments but may be important contributors to development or morphology. For instance, the absence of these proteins may explain EYP’s greater pH transition when titrated with NaOH (Figure 3B). To benchmark differences between EY and EYP cell culture potential, our newly developed 3D cryo well-insert technology [66] could serve as a platform. The 3D well insert uses histology to analyze 2D or 3D cell cultures bypassing in-well microscopy and EY’s light transmission issues. Pre-existing or formed biological contaminants were removed via pasteurization [67,68]. Alternative contaminant ridding methods such as heat sterilization were tested but caused protein coagulation [26,69] and syringe filters (0.2 or 0.45μm) rapidly clogged.

The question of variability has been a large debate in the development of this biomaterial. For instance, some scientists have expressed their belief that the egg biomaterial is too complex. On the other hand, others believed that one egg vendor was not sufficient. We took into account both perspectives as much as possible. To control reproducibility to a certain extent, the eggs were purchased from the same vendor and subtype. Over the course of the experiments, at least 15 egg dozens were purchased from the same vendor. Importantly, each egg dozen did not come from the same farm. Furthermore, we retrospectively analyzed the egg box barcodes and noted that some boxes were sourced from a different province. We believe sourcing our eggs from different farms should capture, to a certain extent, a larger sample pool. We have also been able to reproduce our EYP experiments in another country (Finland) under the same centrifugation conditions, and a similar cell culture experiment showed survival to 14 days [70]. This at least suggests that the production of EYP and its cell-permissive environment is not specific to the company President’s Choice’s eggs.

For cell survival, the Live/Dead quantitative experiments were selected, since our pilot studies showed that EYP without cells would saturate MTT^®^ and Alamar Blue^®^ assays (Data not shown). The EY—initially a cell with a large deutoplasm [71]—most likely contains elements that saturate these metabolic assays. In most experiments, the cells were at the base of the dish; therefore, we believed that no benefits would arise from using 3D confocal microscopy. Our data are very representative of the cells in the culture environment, since the entire surface area of each well was quantified rather than a few selected high magnification images. EYP + Medium permitted cell survival unlike EYP + PBS at the same at 70:30 ratio. Epimax medium’s higher 7.6 pH or higher concentration of weak base (sodium bicarbonate) could explain Epimax’s pH modifying ability over PBS. EYP’s pH appeared to be one of the main contributing factors of cell death. This was further demonstrated when cells survived in medium-free EYP supplemented with either NaOH or EW.

To alleviate EYP’s dehydration—seemingly caused by ambient air—PBS was added to EYP + NaOH (EYP7.4) and improved survival with dilutions 30–70%. Consequently, these dilutions diluted EYP’s content. To maintain the highest EYP content and hydration, future studies should consider exchanging the PBS dilutions with varying levels of air humidity. This is often discussed in avian embryonic developmental studies [72,73]. In control groups, PBS and PBS + BSA cell survival was not observed. In the PBS control group, the cell death could be attributed to PBS’s lack of nutrition since pH (Figure A6) and osmolality (Figure A8) criteria were met. Cell death in BSA (PBS + BSA) went against our expectations. BSA mixture’s pH was within range (Figure A6) while osmolality values of PBS—the solvent—should not drastically change since in human serum, proteins only affect osmolality by 0.5% since they are in mmol concentration [74]. The EYP appears to contain particular elements absent from the BSA that promote cell survival and proliferation. This observation correlates with Marakumi et al.’s YLP-pI7.5 egg-derived growth-promoting factor [43]. Future studies could investigate the egg biomaterials’ cell survival permissive factors. Experiments were also conducted to evaluate our egg biomaterials’ osmolality (Figure A8). EW at 238 mOsmol/Kg (S.E 3.5) and EpiMax medium at 324 mOsmol/Kg (S.E 0.8) were respectively at the low and high ranges and near the physiological osmolality range (260–320 mOsm/kg) [75]. These experiments show that our egg mixtures did not critically shift osmolality and were within the physiological range further supporting their relevance as a tissue engineering biomaterial.

In salivary glands, the intralobular stroma distributes the fibroblasts, while acinar cells are clustered into acini [76]. In vitro, in the EYP + Medium, both cell types displayed different morphologies (Figure A4). HuSG-Fibro had greater affinity to the culture plate’s surface while the NS-SV-ACs preferentially attached to themselves. EYP either prevents strong attachment of the NS-SV-ACs or promotes their intercellular connection. The Live/Dead chemical cocktail, when added, also contributed mainly to the detachment of NS-SV-AC cells (Figure A9B). In bright-field microscopy, only the HuSG-Fibro formed non-cellular translucent halos around their anchoring location (Figure 2M dotted line). Cell culture medium containing “crowders” have shown their ability to enhance cell collagen production by the excluding volume effect [64,65,77]. The translucent halo could potentially correspond to secreted human ECM. As the HuSG-Fibro was less proliferative, future studies should investigate if the “crowders” in the egg biomaterial shift the cells’ potential to production rather than proliferation. Furthermore, with regard to both cell types’ proliferation, it is important for the reader to consider that cell proliferation is not always a desired aspect of cell culture since in vivo, cells are not always proliferating. Sediments were also observed under certain circumstances in bright field microscopy. They were negative to Live/Dead stain excluding them as bacteria. On the rare occasion of bacterial infection, the Ethiudum homodimer DNA binding dye clearly signaled the bacteria’s presence. Pasteurization inaccuracies could have caused a minority of proteins to denature or coagulate, resulting in sediments. The addition of Live/Dead cocktail to the wells helped re-suspend the sediments for clearer imaging (Figure A9A). Cell cytoplasm appeared darker in bright light microscopy with more EYP (Figure A4 and Figure A5). The accumulation of lipids droplets has been shown elsewhere in EY-supplemented cultures. The droplets were made visible with Oil O Red staining [42]. Specific or non-specific endocytosis could be used by the cells to capture extracellular molecules from the egg biomaterial cultures [78,79]. The changes in the cells’ cytoplasmic transmittance of light and their survival and proliferation without medium suggest that endocytosis occurs and feeds the cells. In egg biomaterials, the human cells’ lipid metabolism remains uncertain; future studies should specifically investigate human cells’ ingestion and use of egg components.

The EYP + EW biomaterial combination outperformed our expectations. In avian development, EY and EW juxtaposition has a purpose; at their junction, the blastoderm develops at pH 7 [30]. Furthermore, EW transfers water to the EY [80]. Similar to the developing embryo, our mixture of EYP + EW 50:50 neutralized EYP’s pH and brought necessary hydration. Most importantly in EYP + EW, cells grew/survived suspended in a 3D conformation without commercial medium or supplements (Figure 5 and Appendix A). Alternatively, EYP and EYP7.4 can both gel [70]. Death observed in the weaker NS-SV-AC cultures could be attributed to the mechanical forces used to mix the Live/Dead dyes into EW gel opposing the more robust primary fibroblast. At an EYP + EW 50:50 ratio, the EW acted in three ways: (1) it modified the EYP’s pH, (2) its gelatinous structure permitted greater 3D distribution of the cells and (3) the water content in the EW protected the EYP from dehydration. 

In foreshadowing clinical implementation, we understand that xenocontamination is a challenge for certain biomaterials and is not exclusive to the egg. For instance, collagen biomaterials have been modified to decrease immunoreactivity (atelocollagen) [77], and an analogous approach may be used for the egg biomaterial. Alternatively, transplant patients after their surgery are placed on immunosuppressive drugs. We expect the same may be needed from our implantation. More recently, egg whites have also been used as a sponge for mice in vivo experiments showing low immune rejection [81]. Our vision of the EYP (and egg-derived materials) is that it can be used as a bio-reactor rather than an implantable biomaterial. For instance, after feeding on the EYP, using the EW’s network for 3D distribution and producing human ECM, the cells—or tissues at this point—could be removed from the egg environment, washed, and clinically implanted.

## 5. Conclusions

Our focus was to discover the requirements for human salivary glands cell survival in the translucent EYP. This was important since eggs are inexpensive, abundant, and developmentally encouraging biomaterials that could potentially act as standalone tissue engineering incubators. On its own, EYP was not conducive for human cell survival. Alternatively, EYP + Medium, EYP + NaOH and EYP + EW can serve as biomaterials for human cell culture. Most importantly, for cells to survive in EYP, EYP needs a pH modification and sufficient hydration. NaOH sufficiently changed EYP’s pH to permit cell survival without commercial cell culture medium. EW sufficiently changed EYP’s pH, provided hydration, 3D distribution, and permitted cell survival without cell culture medium. To our knowledge, this is the first study to demonstrate the growth and/or survival of human cells in non-fertilized egg EYP components in vitro. Scientists can use these EYP biomaterials in tissue engineering or for drug-screening applications.

## Figures and Tables

**Figure 1 materials-13-04807-f001:**
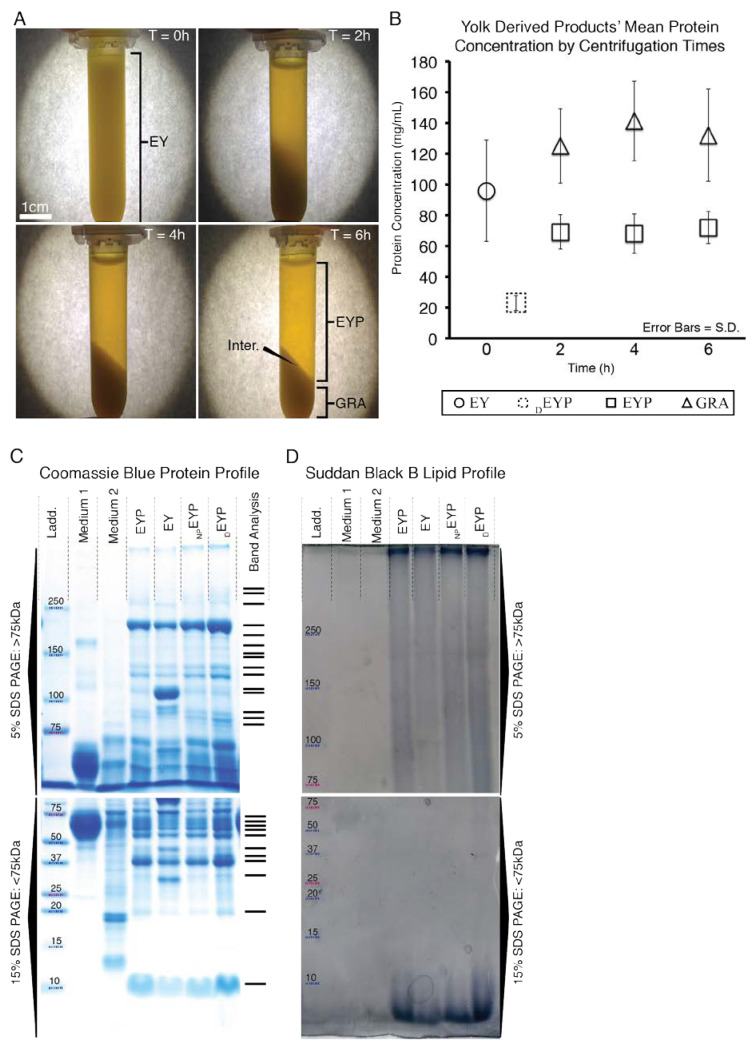
Images and Quantative Values of EYP During Centrifugation and Images of EYP, EY and Culture Media SDS-Pages Stained with Coomassie and Sudan Black B. (**A**) Images of 2 mL tubes after centrifugation and held in front of a lit piece of paper. After 6 h of spinning, the egg yolk (EY) becomes the egg yolk plasma (EYP) and granules (GRA) separated by an interface (Inter.) (**B**) Protein values of EY, EYP & GRA fractions as well as in Diluted EYP (_D_EYP) over centrifugation times as quantified by Bicinconinic Acid assay. (**C**,**D**) The SDS-PAGESs 5% and 15% compare EYP to both used cell culture media: RPMI 1640 containing 10% FBS (Medium 1), EpiMax (Medium 2) and to EY, non-pasteurized EYP (_NP_EYP) and _D_EYP. The protein standard ladder (Ladd.) is identical in all SDS-PAGEs. (**C**) A digital scan of SDS-PAGEs stained with Coomassie Blue (appearing as blue bands). Band Analysis indicates software detected peaks in only EYP profile. (**D**) A digital scan of SDS-PAGE stained with Suddan Black B comparing quantity of lipids (appearing as dark shadows).

**Figure 2 materials-13-04807-f002:**
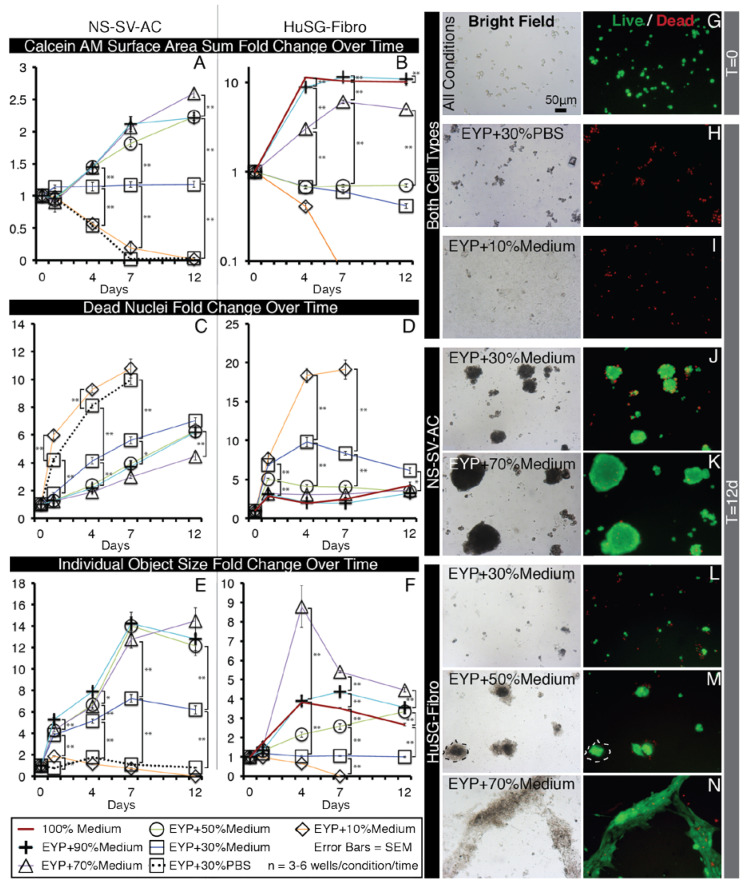
Selected Bright Field and Fluorescent Images of NS-SV-AC and HuSG-Fibro Cells in EYP Cultures with Different Concentrations of Medium, Stained with Live/Dead and Images Quantified by Image Analysis. (**A**–**F**) Whole well images from 96 well plates were captured with 2X objective and subjected to computer analysis algorithms to detect Calcein AM (live) areas (in μm^2^) and EthD (dead) nuclei counts then normalized to T = 0. (**A**,**B**) In sums, green pixels were added in each well. (**C**,**D**) For dead cells, individual dead nuclei were counted in each well. (**E**,**F**) With objects, pixel values for either individual cells or clusters of cells were considered an object and means of their area were calculated per well. For statistical analysis we used one-way ANOVA followed by Tukey’s Post-Hoc test with *p*-values = * <0.05, ** <0.01. (**G**–**N**) Images captured with 10X objectives from inside wells or a 96 well plate shown representative morphology of Live Dead stained NS-SV-AC or HuSG-Fibro cells in various conditions. Dotted line (**M**) indicates HuSG-Fibro suspected produced substance. More images are found in Figure A4.

**Figure 3 materials-13-04807-f003:**
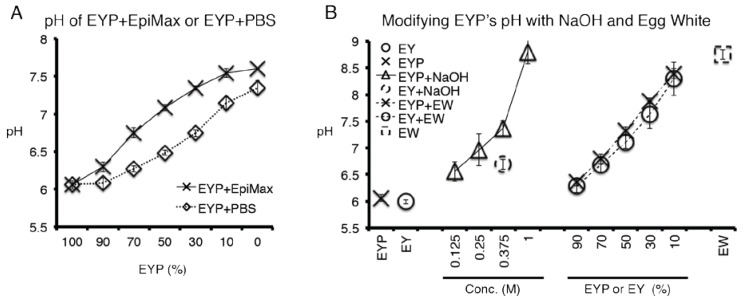
Graphs Showing EYP’s pH Change When Combined with Medium, NaOH and Egg Whites. (**A**) EYP diluted with different concentrations of Epimax medium and PBS. (**B**) NaOH and EW’s effect on EYP and Egg Yolk’s (EY) pH. In these experiments, all tubes contained 5 mL of EYP or EY. Then for the Conc. experiment, 0.25 mL of the shown concentration was added to 5 mL of either EYP or EY. For the EW experiment, EW quantities were added to the initial volume of 5 mL, increasing EW percentage. All pH measurements were conducted at 37 °C with a pH meter and probe. Error bars are standard deviation.

**Figure 4 materials-13-04807-f004:**
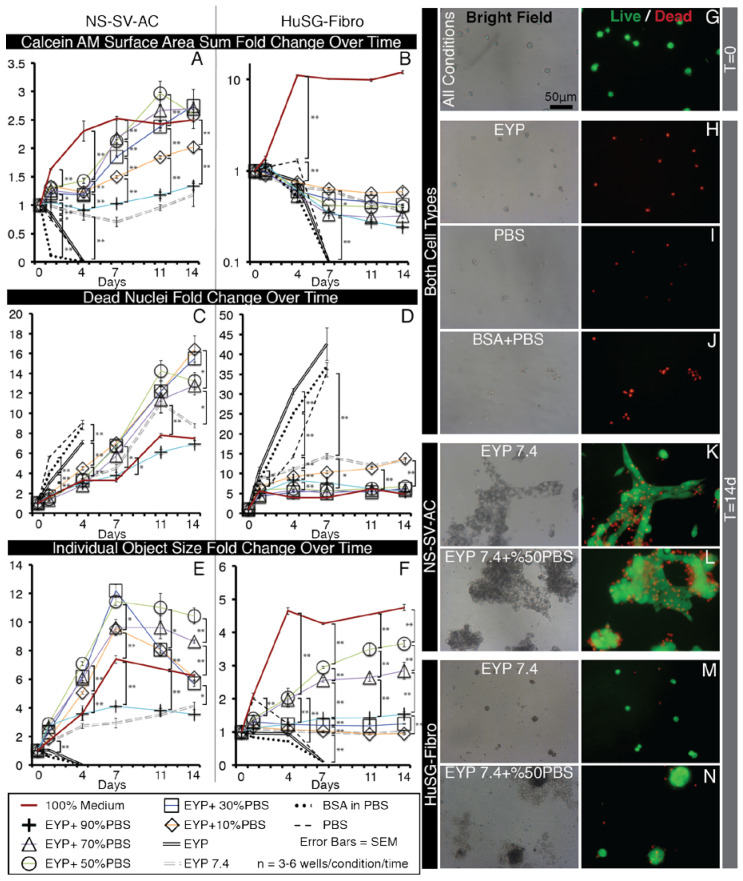
Selected Bright Field and Fluorescent Images of NS-SV-AC and HuSG-Fibro Cells in EYP + NaOH (EYP7.4) Cultures with Different Concentrations of PBS, Stained with Live/Dead and Images Quantified by Image Analysis. (**A**–**F**) Whole well images captured with 2X objective were subjected to computer analysis algorithms to detect Calcein Am (Live) areas (in μm^2^) and EthD (Dead) nuclei counts then normalized to T = 0. (**A**,**B**) In sums, totals of green pixels were added in each well. (**C**,**D**) For dead cells, individual dead nuclei were counted in each well. (**E**,**F**) With objects, pixel values for either individual cells or clusters of cells were considered an object and means of their area were calculated per well. For statistical analysis we used one-way ANOVA followed by Tukey’s Post-Hoc test with *p*-values = * <0.05, ** <0.01. (**G**–**N**) Images captured with 20X objectives from inside wells of a 96 well plate shown representative morphology of Live Dead stained NS-SV-AC or HuSG-Fibro cells in various conditions. More images are found in Figure A5.

**Figure 5 materials-13-04807-f005:**
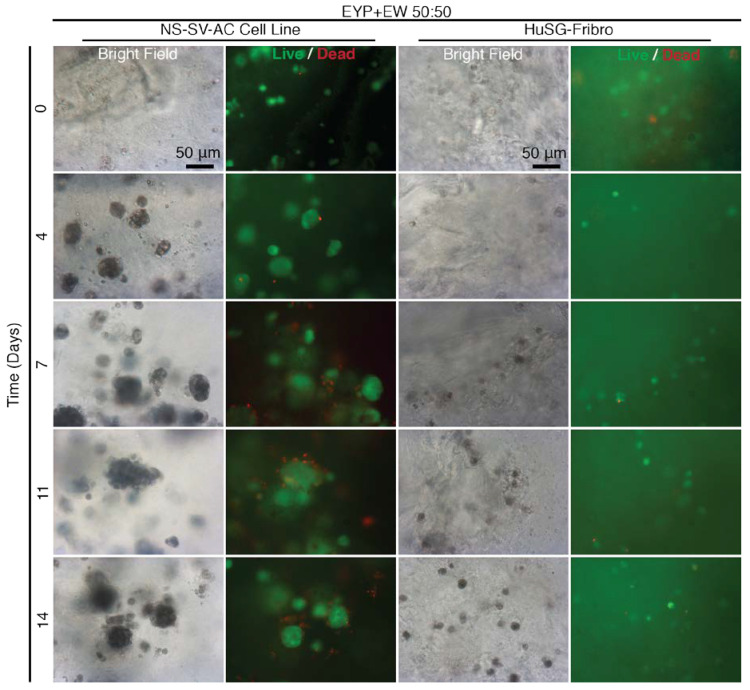
Bright Field and Fluorescent Images of NS-SV-AC and HuSG-Fibro Cells in EYP + EW 50:50 mixtures stained with Live/Dead over 14 days. Images were captured with 20X objective at a random plane of focus in the well. Scale bars are identical in all images. Bright field movies showing progress through the z layers can be found in Appendix A.

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
