# Peer review of "3D Cell Culture of Human Salivary Glands Using Nature-Inspired Functional Biomaterials: The Egg Yolk Plasma and Egg White"

_materials, 2020, doi:10.3390/ma13214807_

Round 1

Reviewer 1 Report

This study by Charbonneau et al. describes a new cell culture system based on egg yolk plasma (EYP), which consist of high concentrations of lipid (mostly LDL) and protein. The results indicate that EYP could be an inexpensive and highly effective substitute for more commonly used bovine serum for cell culture studies.

  1. Is there a reason for specifying human salivary gland cells? This work would be much more significant if EYP-based culture could be applied to a wide variety of cells.
  2. It would significantly enhance the quality of this result if the cell culture studies were conducted using conventional cell culture media containing fetal bovine serum as a control.
  3. What are the proteins in EYP as determined by SDS-PAGE? The authors indicated several bands, but did not identify any of them. Even if it is not possible to identify all of them, at least key proteins that are vital for cell culture should be identified.

Reviewer 2 Report

In this manuscript by Charbonneau et al., the authors have analytically tested the effect of egg yolk plasma pH on cytotoxicity and demonstrated the combined use of commercial culture media or egg white with egg yolk plasma as a biomaterial for salivary gland cell culture. Overall, the manuscript is well-written and the experiments are carefully designed. However, from a reader’s standpoint, the data representation is clumsy, which when improved will make the manuscript more attractive and easier to comprehend. Concerns regarding the data interpretation and representation are written below:

  1. Page 7, line 223: “we detected 26 bands; 15 above….”. What is the implication of these bands? What do these represent? Authors must clearly state.
  2. In section 3.3 and 3.4, the authors have conducted identical experiments for two different cell lines. Can the authors simplify this section by drafting a single section and comparing the effect of media and other stimulants on both cell lines side by side? This comparison will make it easier (and faster) for the readers to comprehend the key findings.
  3. Some figures have too many panels which can be reduced to convey only the key information to the readers. For instance, the authors can show 1, 50 and 90% media in the main file and add the other concentrations to the supporting information as these concentrations are sufficient for comprehend the authors’ findings. Also, can the authors color-code the experimental groups of quantitative data in all figures for the reader’s ease? In some panels, e.g. fig 3c, 5 b, c the symbols are overlapping, making it difficult to relate them with corresponding experimental groups.
  4. In some figures e.g. fig 7, a qualitative data can be shifted to the supporting information while the main manuscript file only displays meaningful quantitative data.

Based on the current version of the manuscript, a revision is recommended.

Reviewer 3 Report

As with any research using food produce as a starting material, one of the major concerns is the inclusion of the additives. While this work is rather fascinating on its own account and relatively noteworthy, the reviewers feels that it is necessary to address the concerns pertaining to the source of egg. Certainly, a “Large White Eggs Omega 3, Presidents’ Choice” sample pool is not a representative of all eggs available in the market.  Overall, writing is generally poor and at times felt as it had been translated via software. The reviewer understands that is an extremely harsh comment but this is the impression that had been given to the reviewer.  For example, “For the moment, the medium’s “reduced” content with or without serum can satisfy cell culture’s requirements. If human sized tissues will require highly dense and nutritional biomaterials, the contents of eggs could act as a low cost, 3D matrix environment similar to an egg”.  This sort of poor writing is extremely obvious in the discussion section.  Furthermore, all fluorescence images are very blurry and lacked discernable features that would make sense whatsoever. And interestingly, the insertion of supplementary figures before the main figures had somewhat render this manuscript rather messy at times and difficult to follow.  The reviewer wonders if this is a result of the manuscript editing during submission or of deliberation from the authors part. Figure captions are strange and as it typically initiates with a single of data description rather than telling the reader what the set of figures represent right at the onset. This is rather confusing to the reviewer. Borders from the graph insets seems to suggest that the data presentation is a hastily “cut and paste” job from somewhere and overall, the data presentation is extremely poor. Furthermore, there is no notable graphical schematic of what had been done in this work.

Overall, this manuscript reads more like a lab book rather than a proper academic discourse while the selection of a single commercial egg distributor is rather unscientific. The considerations on other types of egg sources etc should be addressed adequately.  On the basis of all these issues, the reviewer cannot see anything other than a rejection at stage.

Reviewer 4 Report

Please add a space before references

Several sentences lack references that must be added. See lines 33, 44, 60…

Lines 52-61: I suggest this information be moved to the discussion section since it discusses the cells selection and the methodology used

Lines 85-95: this paragraph needs to be rewritten. I suggest the authors provide a more clear and focused study hypothesize. Also, the authors provide here a methods and results summary, that I understand was to justify the second hypothesize; however, I suggest the redundant information be removed, and other moved to the discussion section

Line 99: what subspecies of chicken provided the eggs? Gallus gallus domesticus? This is important because variation on egg content could exist between different chicken subspecies

Lines 105-107: since the EY were put in an “70% ethanol sprayed towel” do authors think this can cause some proteins to denaturate? The authors discuss the sediments present, which they attribute to pasteurization inaccuracies; however, this step of placing the EY in ethanol sprayed towel could also lead to denaturation of more superficial proteins in the EY.

Line 112: how were the centrifugation conditions chosen/determined?

Line 126: it is not clear in which step this centrifugation times occurred. I believe it is during isolation and preparation steps. However, in this step, the authors only refer to a centrifugation time of 6 hours. Please explain what T0, 2, 4, and 6 mean here.

Section 2.4: several information needs to be added to this section related to the use of the human samples. Please add ethical approval, patient characterization, how the human salivary tissues were collected…

Line 165: what cell passages were used for experiments? Please add this information to the manuscript

Line 166: please add the cellular density used for experiments

Line 183: please add the time in hours of the “longer incubation period”

Section 2.8: did the authors performed a normality test to choose the appropriate statistical test to be used?

Lines 232-233: the authors stated that “EYP on its own was not conducive to cell survival.” However, no results are seen in the figure for EYP alone. I suggest adding the data or stat “data not shown” on the manuscript

Line 247: I believe “were observed” is missing on the sentence

Please add the main study limitations to the discussion section

The authors refer to egg advantages as stand-alone tissue engineering incubators for human salivary gland cell expansion. However, human cell expansion for clinical use is subject to several restrictions, and xenocontamination is a significant issue. Please discuss this.

Lines 442-445: please fill in the author's contributions

Supplementary figure 1: I think it will be interesting to see statistical comparisons between EYP and the groups were EpiMax was mixed with EYP, to better understand the threshold were EpiMax addition promotes a significant decrease in protein concentration

I found the supplementary figure 2 unnecessary, primarily because most of the figures have already been published. I suggest the appropriate reference be made in the text instead.

Figures S3, S5, S7, 1, 2, 3, 4, 5, 6, and 7: figures present uppercase letters (A,B,C..), while the figures captions present lowercase letters (a,b,c..). Please uniformize.

Figure S5 graphs: please add symbols to indicate statistically significant differences when appropriate.

Figure S6 graphs: please add symbols to indicate statistically significant differences when appropriate.

Figure S1 graph: please add symbols to indicate statistically significant differences when appropriated

I do not agree with section 3.7 and figure 7 reference to spheroids 3D cultures. I understand the obtained images of the clustered cells are suggestive of a spheroid form, but only 2D images are shown, which do not allow us to confirm the spheroid form clearly.

Round 2

Reviewer 1 Report

The revised manuscript shows much needed improvement for publication.

Author Response

We thank the reviewers for their time on the second revision. Since the last submission, minor modifications were made on the manuscript. Mainly the abstract was revised, the graphical abstract was completely reviewed and throughout the text, minor sentence fragments were corrected to improve clarity and grammar.

Reviewer 2 Report

The authors have addressed my comments. Hence, I recommend the revised manuscript for the publication in “Materials”.

Author Response

(The authors gave the same response as above.)

Reviewer 3 Report

The reviewer appreciate the effort undertaken by the authors and would like to recommend a minor revision as the graphical abstract remains unclear in the reviewer opinion. Please revamp the graphical abstract as it is rather different to understand at the first glimpse without any figure captions.

Author Response

(The authors gave the same response as above.)
